# Antimicrobial and Immunomodulatory Effects of Selected Chemokine and Antimicrobial Peptide on Cytokine Profile during *Salmonella* Typhimurium Infection in Mouse

**DOI:** 10.3390/antibiotics11050607

**Published:** 2022-04-30

**Authors:** Astrid Tuxpan-Pérez, Marco Antonio Ibarra-Valencia, Blanca Elisa Estrada, Herlinda Clement, Ligia Luz Corrales-García, Gerardo Pavel Espino-Solis, Gerardo Corzo

**Affiliations:** 1Departamento de Medicina Molecular y Bioprocesos, Instituto de Biotecnología, Universidad Nacional Autónoma de Mexico, A.P. 510-3, Cuernavaca 62250, Mexico; herlinda.clement@ibt.unam.mx (H.C.); ligia.corrales@udea.edu.co (L.L.C.-G.); 2Translational Research Laboratory, National Laboratory of Flow Cytometry, Autonomous University of Chihuahua, Circuito Universitario, Campus II, Chihuahua 31109, Mexico; blancaelisaestrada@gmail.com (B.E.E.); gespinos@uach.mx (G.P.E.-S.); 3Departamento de Alimentos, Facultad de Ciencias Farmacéuticas y Alimentarias, Universidad de Antoquia, AA 1226, Medellín 050010, Colombia

**Keywords:** antimicrobial, cytokines, inflammation, peptides, *Salmonella enterica* serovar Typhimurium

## Abstract

The antimicrobial and immunomodulatory capacities of the peptide Css54 and the chemokine MCP-1 were tested. The first, a peptide isolated from the venom of the scorpion *Centruroides suffusus suffusus* was synthesized chemically. In contrast, the second is a monocyte chemoattractant expressed as a recombinant protein in our lab. It was observed in vitro that Css54 inhibited the growth of *Salmonella enterica* serovar Typhimurium (6.2 µg/mL). At high concentrations, it was toxic to macrophages (25 µg/mL), activated macrophage phagocytosis (1.5 µg/mL), and bound *Salmonella* LPS (3 µg/mL). On the other hand, the recombinant MCP-1 neither inhibited the growth of *Salmonella* Typhimurium nor was it toxic to macrophages (up to 25 µg/mL), nor activated macrophage phagocytosis or bound *Salmonella* LPS (up to 3 µg/mL). Although it was observed in vivo in mice Balb/C that both Css54 and MCP-1 did not resolve the intraperitoneal infection by *S.* Typhimurium, Css54 decreased the expression of IL-6 and increased IL-10, IL-12p70, and TNF-α levels; meanwhile, MCP-1 decreased the expression of IFN-γ and increased IL-12p70 and TNF-α. It was also observed that the combination of both molecules Css54 and MCP-1 increased the expression of IL-10 and TNF-α.

## 1. Introduction

Sepsis is a clinical syndrome resulting from an unregulated inflammatory response to microbial infections. It involves the dysfunction of vital organs such as the heart, lungs, and kidneys. Severe sepsis and septic shock are the main causes of death in nosocomial patients [1]. According to the World Health Organization, 11 million people die of sepsis each year [2]. In addition, globally, 85% of cases and 84.8% of deaths related to sepsis occur in countries with a low, medium-low, or medium sociodemographic index [2].

The immune response observed in sepsis occurs in an early phase of immune hyperactivation, and a late chronic phase called persistent immunosuppression. The early phase causes widespread inflammation with multiorgan failure and a phenomenon known as the cytokine storm, where a large number of pro-inflammatory cytokines are released into the bloodstream—mainly TNF-α, IL-1, and IL-6 [3,4]. Meanwhile, during persistent immunosuppression, the death of the individual may occur due to a hyper-inflammatory response. The main cytokines expressed in this phase are IL-6, IL-12p70, IL-17, and IL-8 [5,6].

*Salmonella enterica* serovar Typhimurium, *S.* Typhimurium, is a Gram-negative bacillus whose infection occurs from contaminated water or food ingestion. It crosses the intestinal mucosa in the intestine and adheres to epithelial cells. In mice, this occurs preferentially in M cells of Peyer’s patches [7]. After adhesion, *S.* Typhimurium modifies the epithelial cytoskeleton of these cells to enter through *Salmonella*-containing vacuoles (SCVs). Simultaneously, the recruitment and migration of phagocytes to the submucosal space of the intestinal lumen begin; this process is associated with the production of pro-inflammatory cytokines such as TNF-α and IL-8 [8]. Finally, bacteria spread through the lymphoid nodes and eventually through organs such as the liver and spleen [9].

Multicellular organisms have developed small peptides called ‘host defense peptides’ (HDP) as part of their innate immune system that target pathogens such as bacteria, viruses, and parasites [10]. These peptides are short chains with 10 to 50 amino acids, positively charged (generally from +2 to +9) with a significant portion of hydrophobic residues (greater than or equal to 30%) [11]. Hydrophobicity, cationicity, and secondary structure are implicated in its antimicrobial effects [12].

In 2013, Garcia et al. [13] isolated from the venom of the scorpion *Centruroides suffusus suffusus* a peptide of 25 amino acid residues, named ‘Css54’. This peptide acquires an alpha-helix structure in hydrophobic environments due to its large number of aliphatic and hydrophilic residues and lacks disulfide bridges. It is hemolytic and it has antimicrobial activity against Gram-positive and Gram-negative organisms such as *Staphylococcus aureus* and *Escherichia coli*, at a minimum inhibitory concentration of 12.5 µg/mL [13]. Css54 was also tested in vitro combined with anti-tuberculosis antibiotics such as ethambutol, pyrazinamide, isoniazid, and rifampicin. It was observed that it had a synergistic effect with those antibiotics [13].

Alternatively, the monocyte chemoattractant protein 1 (MCP-1), also known as ‘CCL2’, belongs to the CC chemokine family and has 76 amino acid residues. It exhibits a chemotactic activity toward monocytes, T cells, and dendritic cells; and promotes IL-4 secretion, an anti-inflammatory cytokine, and LTB_4_, a neutrophil attractant [14,15,16]. MCP-1 is secreted in response to various stimuli, including infectious agents and inflammatory cytokines [17]. Some reports have mentioned the importance of MCP-1 in resolving wound infections and sepsis [18,19], despite its lack of antimicrobial activity. This effect is often attributed to its influence in innate and adaptative immunity through the activation of monocytes and T helper cell polarization, respectively [14,15].

Thus, our interest arises in examining whether the antimicrobial capacity of the Css54 peptide and the immunomodulatory capacity of the MCP-1 chemokine are capable of resolving a sepsis process caused by *Salmonella* Typhimurium in a murine model. Either Css54 or MCP-1 as individual treatments or in combination.

## 2. Results

### 2.1. Purification of Css54 Peptide and MCP-1 Chemokine

The crude synthetic peptide Css54 was purified by *Reverse phase high-performance liquid chromatography* (RP-HPLC) with a C_18_ column as a peptide of synthetic origin. The peptide was eluted at a concentration of 47% acetonitrile +0.1% TFA (Appendix A). The main fraction collected was analyzed by mass spectrometry (MS), where molecular mass of 2870.1 Da was obtained, corresponding to the theoretical molecular mass of Css54 (2870.4 Da) (Appendix A). Alternatively, MCP-1 was expressed in *E. coli* Origami cells. The protein was found in both soluble and insoluble fractions, but MCP-1 was purified from the soluble fraction by affinity chromatography. Although a molecular mass of 10,935.4 Da was expected, including the histidine tag, a band with an apparent molecular weight of 14 kDa was observed by SDS-PAGE (Appendix A). This may be due to numerous positive charges (+14) and its basic isoelectric point (9.36), which results in an anomalous run in SDS-PAGE [20].

The fractions obtained by affinity chromatography were subjected to two purification steps by RP-HPLC using a C_4_ column. The protein was eluted at a concentration of 32.8% acetonitrile (Appendix A). The collected fraction was analyzed by MS, with a molecular mass of 10,929.7 Da. The theoretical molecular mass of the oxidized peptide is 10,931.4 Da, with the histidine tag; therefore, the oxidized form was obtained. The 1.7 Da difference could be related to equipment error (Appendix A).

### 2.2. Css54 Is Bactericidal against Salmonella Typhimurium

Previously, Garcia et al. (2013) [13] demonstrated that Css54 inhibits the growth of *S. aureus* and *E. coli* at 12.5 µg/mL. For this study, concentrations from 0.7 to 100 µg/mL of Css54 and MCP-1 against *S.* Typhimurium ATCC 14028 were tested to determine its antimicrobial activity by serial microdilution assays. Css54 had a minimal inhibitory concentration (MIC) of 6.25 μg/mL (Figure 1). However, MCP-1 did not have antibacterial activity, although a decrease in growth was observed regarding the control at 50 µg/mL concentration, but with no statistical significance. Furthermore, the minimum bactericidal concentration (MBC) of the Css54 peptide against *S.* Typhimurium was 25 µg/mL. Therefore, according to the French et al. (2006) criteria, which establishes that the minimum concentration of a peptide that prevents bacterial growth represents the MBC if it is less than four times the MIC [21], we concluded that Css54 had a bactericidal effect on *S.* Typhimurium.

### 2.3. Css54 Is Cytotoxic against Macrophages

Highly hydrophobic defense peptides are cytotoxic for eukaryotic cells, which is why the effect of Css54 and MCP-1 on the viability of RAW 264.7 cells was determined. Concentrations from 3.1 to 100 μg/mL of Css54 and MCP-1 were tested. Our results indicate that Css54 was toxic to cells from concentrations greater than 12.5 μg/mL, but MCP-1 was not (Figure 2). These results permitted the in vitro experiments to continue using a maximum concentration of 12.5 μg/mL of both Css54 and MCP-1.

### 2.4. Css54 Promotes Phagocytic Activity of Macrophages

Since phagocytosis is one of the main activities of macrophages when they contact a pathogen, it was examined whether Css54 or MCP-1 influence this type of activity in RAW 264.7 cells. Different concentrations (from 0.78 to 12.5 μg/mL) of the Css54 and MCP-1 were tested. An increase in the phagocytic activity of *S.* Typhimurium with Css54 was observed in the control (Figure 3), starting with 1.5 μg/mL, 6.25 μg/mL, and at higher concentrations, such as 12.5 μg/mL. However, as shown in Figure 4, MCP-1 had a weak effect on the phagocytic activity at 1.5 µg/mL, although it was not significant.

### 2.5. Css54 Binds to the Bacterial Lipopolysaccharide

Limulus Amebocyte Lysate (LAL) assays were performed to measure the ability of Css54 and MCP-1 to neutralize lipopolysaccharide (LPS) from *S.* Typhimurium since LPS is one of the main inflammation-causing agents. Endotoxin levels (EU/mL) are shown, and Css54 neutralizes LPS in a dose-dependent manner (Figure 5A). There was 100% inhibition of LPS by Css54 from a concentration of 3 μg/mL. However, it showed activity at concentrations as low as 0.18 μg/mL (with 17.8% inhibition), suggesting a high affinity of Css54 for LPS (Figure 5B). A similar test was performed to determine whether MCP-1 neutralizes LPS. In Figure 6, it is shown that MCP-1 slightly reduces the levels of endotoxin concerning the control at 6 μg/mL (Figure 6A). Figure 6B shows that the maximum percentage of LPS inhibition with MCP-1 was 44% at 6 µg/mL.

### 2.6. Css54 Increases Bacterial Load in the Murine Model of Sepsis

Mice groups were treated with a single dose administration of 15 µM of Css54, MCP-1, or Css54 + MCP-1 after 24 h of infection. Mice weight in all groups and evolution of the infection was monitored. It was observed that the treatment with Css54 or MCP-1 or in combination did not prevent mice from losing weight (*p* < 0.0001). Weight loss was associated with the infection and not with the treatments since the control groups maintained their body mass (Figure 7).

After administering the treatments, it was verified whether there was a reduction in bacterial load in the peritoneum, liver, and spleen. Results showed that there was a slight reduction of the bacterial load in the group treated with the Css54 peptide in the peritoneum. However, it was not significant. Treatment with MCP-1 increased the load, and the combination of the peptide and the chemokine left it at the same levels as the untreated group. Alternatively, at the liver and spleen levels, an increase in bacterial load was observed in all cases (Figure 8).

### 2.7. Cytokine Expression Analysis in Murine Sepsis Model

The cytokine expression profiles in the plasma of mice that received 15 μM of Css54, MCP-1, or Css54 + MCP-1 were analyzed to determine the effect that the treatment has on the inflammatory response. As shown in Figure 9, Css54 decreased the expression of IL-6 but increased IL-10, IL-12p70, and TNF-α. Meanwhile, MCP-1 reduced the expression of IFN-γ but increased IL-12p70 and TNF-α. On the other hand, the combination of MCP-1 + Css54 increased the expression of IL-10 and TNF-α. Moreover, in mice that did not undergo an infectious process, there was a change in the cytokine profile expression when MCP-1 was administered, where the expression of all cytokines increased, except IFN-γ. Meanwhile, Css54 increased the expression of IL-6, MCP-1, and TNF-α.

## 3. Discussion

This work determined the antimicrobial and immunomodulatory activity of peptide Css54 and the effect on the immune response of MCP-1 in a model of infection by *Salmonella* Typhimurium. Previously, Garcia et al. (2013) [13] determined that Css54 inhibited the growth of *E. coli* and *S. aureus* at 12.5 µg/mL. Here, we demonstrate that the MIC for *S.* Typhimurium was lower (6.25 µg/mL). The difference in MIC between these bacteria may be due to a particular cell membrane composition. For example, the negative charge of an LPS efficiently interacts with the positive charge of AMPs. This also occurs in the case of Gram-positive LPS that have ionic lipids called teichoic acids [22]. On the other hand, MCP-1 did not present an antimicrobial effect as was reported by Yang et al. (2003) as Gram-negative [23]. This chemokine has also been shown to have no effect against Gram-positive bacteria or parasites such as *S. aureus* and *Leishmania mexicana* [23,24]. Yet, an important property of an AMP is to be considered a defense peptide (HDP) that kills bacteria without being toxic against eukaryotic cells. Here, the results showed that the concentration that Css54 requires to function as an antimicrobial (6.25 µg/mL) is lower than that which is toxic to macrophages (25 µg/mL). This gave us a working range from 25 to 6.25 µg/mL for testing in vivo models. Although, as mentioned earlier, in vitro and in vivo models may present different responses to the molecules tested, just as was observed in the murine model. These differences may be attributed to such molecules’ physicochemical characteristics or the animal model’s response and metabolism. At the same time, MCP-1 is not cytotoxic at any concentration tested, proving its safety for in vivo models.

Macrophages kill bacteria through phagocytosis or by releasing inflammatory molecules and nitric oxide (NO) [25]. According to our results, Css54 increased phagocytic activity in macrophages against *S.* Typhimurium, even at low concentrations. Several reports have shown that HDPs can activate macrophage functions. Cathelicidin LL-37, as an example, increases bacterial phagocytosis by human macrophages at low concentrations [25], sublancin (produced by *B. subtilis*) improves the phagocytic activity against *S. aureus* in vitro and in vivo [26], and HD5 (Human β-defensin 5) facilitates the internalization of *Shigella* and *Salmonella* in macrophages by promoting bacterial adhesion [27]. Alternatively, chemokines can perform different functions depending on the stimulus. CYTL1 promotes phagocytosis of *E. coli* and induces chemotaxis in neutrophils [28]. There is no evidence that CC-type chemokines (as in the case of MCP-1) promote phagocytic activity in macrophages, and in this work, at least with the experiments performed, this lack of activity in the phagocytic activity was corroborated.

Since the treatment of infections with antibiotics promotes the release of pathogenic factors such as LPS, which results in bacterial sepsis, effective treatment should combine strong antimicrobial activity with the ability to bind and neutralize LPS. It has been observed that HDPs can interact with LPS in two ways. First, by the formation of electrostatic bonds between the positive charges of the peptide with the phosphate groups of the negatively charged LPS; and second, by the hydrophobic interaction between them [29]. Our results showed that Css54 neutralized LPS even at concentrations lower than 0.5 µg/mL. This might be due to its large number of positive and hydrophobic residues. It has been suggested that the interaction between peptides and LPS may be due to the binding of cationic residues to the phosphate groups of LPS and the lipid A portion, as occurs with LF12 (derived from Lactoferrin) and Mel4 (derived from Melittin) [30]. Additionally, it has been observed that by replacing hydrophilic residues with hydrophobic amino acids, the ability to bind to LPS increases [31]. Therefore, Css54—having almost 50% hydrophobic residues—has a great capacity for LPS binding. On the other hand, MCP-1 showed a certain ability to neutralize LPS at 6 µg/mL (44% neutralization); this property in chemokines has not been reported. Furthermore, MCP-1 demonstrated an ability to reduce the growth of *S.* Typhimurium in vitro (100 µg/mL), although it did not inhibit it. We might perform further studies for larger periods or increase the chemokine concentration to determine whether these factors improve such effects.

Some AMPs reduce the levels of bacterial growth in organs such as the lungs, liver, and kidneys in murine models, thereby ensuring the survival of individuals [32]. Here, treatment with Css54, MCP-1, or a combination of both, at the concentrations tested failed to reduce the bacterial load in the peritoneum, liver, or spleen, which would also explain the considerable weight loss of mice. Although AMP and chemokine showed interesting biochemical activities in vitro, this is not fully reflected in an in vivo model. This could be due to numerous circulating molecules in the body that interact or aggregate with them, and such molecules being degraded during their distribution in the animal body. A way to evaluate this could be by implementing serum kinetics or analyzing their biodistribution in a murine model. These data may be important for designing better treatments by modifying the dose of peptides and proteins or by using excipients during administration.

Some treatments for sepsis have focused on the neutralization of one or more inflammatory mediators, such as IL-1 receptor antagonists and anti-TNF-α antibodies. However, they have not been entirely effective because anti-inflammatory therapies could increase secondary infections and mortality [33]. Therefore, modulation of the immune system with HDPs or chemokines could be a therapy with great potential for bacterial infections. Here, we analyzed the effect of Css54 and MCP-1 in the profile expression of the inflammatory cytokines IFN-γ, IL-6, IL-12p70, and TNF-α; the anti-inflammatory cytokine IL-10; and the chemokine MCP-1. These molecules play a physiological role during the infection by *S.* Typhimurium. It is worth mentioning that this bacterium also takes advantage of the liberation of pro- and anti-inflammatory cytokines in mammals to cause damage in the intestine and, simultaneously, to evade the immune response [34]. Css54 demonstrated the ability to reduce IL-6 levels and increase the anti-inflammatory cytokine IL-10. Furthermore, it slightly increased the expression of pro-inflammatory cytokines IL-12p70 and TNF-α, which are involved in the inflammatory response and are associated with IFN-γ expression. However, the IFN-γ increase was not enough to be seen as significant, but it certainly has merits for further studies. Curiously, even though MCP-1 did not act as an antimicrobial, it increased the expression of IL-12p70 and TNF-α while decreasing IFN-γ. There was also a non-significant increase in the expression of IL-10 by this chemokine, and a non-significant reduction of IL-6, which it may grant to consideration in further experiments by ELISA that could be helpful to clarify this modulation in the liberation of cytokines. It is worth mentioning that when Css54 and MCP-1 were applied in combination, there was a clear increase in the expression of IL-10 and TNF-α while other cytokines remained at the same level that those presented in the infection group. These data indirectly suggest a tendency to keep control of the inflammatory response and create an anti-inflammatory environment; however, this aspect must be corroborated with the screening of more cytokines, inflammatory molecules, and cell response differentiation. Some HDPs such as LL-37 and Ps-K18 have neutralized LPS and reduced levels of pro-inflammatory cytokines. Ps-K18, a 2702 Da peptide, and a net charge of +3 blocks the activation of TLR4/NF-κΒ and also reduces the production of TNF-α and IL-6 in an in vivo model [35,36]. Css54, having comparable characteristics to Ps-K18, a mass of 2870.4 Da and a net charge of +5, could be acting similarly by modulating the immune response. Some studies have shown that when AMPs interact with LPS suppress cytokine release [37,38]. AP3, an AMP isolated from the scorpion *Trachypithecus obscurus*, decreases the transcription levels of pro-inflammatory cytokines such as TNF-α and IL-1β [38]. Another possible explanation is that Css54 would act as a recruiter of leukocytes to the site of infection, affecting the production of IFN-γ and MCP-1, consequently suppressing the levels of pro-inflammatory cytokines IL-12p70 and TNF-α, which is an effect reported for other HDPs [39], with the particularity that Css54 also increased the synthesis of IL-10.

## 4. Conclusions

Nowadays, therapies against bacterial infection based on AMPs, HDPs, or chemokines are a relevant field of study. Here, we showed that some properties of Css54 and MCP-1 in vitro experiments were lost when these molecules were applied in an in vivo model. However, still, they can be pharmacologically relevant molecules in the inflammatory process. Here, Css54 and MCP-1 were used to resolve a murine sepsis model by *Salmonella* Typhimurium. Still, our results showed that they do not resolve the infection, individually or in combination. Nevertheless, their relevant characteristic relied on their capacity to modify the expression of cytokine profiles when both were used in combination. Therefore, this can eventually be used for developing a suitable therapy, but further experiments must be conducted. For example, other interesting chemokines—such as MIP-1, RANTES, or Fractalkine (C-X3-C motif ligand 1)—may be used to recruit and activate macrophages to increase phagocytic activity.

## 5. Materials and Methods

### 5.1. Bacterial Strains and Cell Line

*Salmonella enterica* serovar Typhimurium strain ATCC 14028 was used, preserved in vials with 30% glycerol at −70 °C. For use, bacteria were streaked onto xylose lysine deoxycholate (XLD) agar or LB broth incubated at 37 °C for 24 h.

RAW 264.7 cell line was used as a cell model. It was cultured at 37 °C with 5% CO_2_ in 100 mm boxes with RPMI medium supplemented with 10% inactivated fetal bovine serum (SFBi); the serum was previously incubated at 56 °C for 30 min. When cells reached 70% confluence, the medium was removed, and passages were performed to maintain cell viability.

### 5.2. Purification of Peptide Css54 and Expression of MCP-1

The synthetic peptide Css54, with sequence FFGSLLSLGSKLLPSVFKLFQRKKE, was manufactured by the GenScript company (Piscataway, NJ, USA). Afterward, the peptide was purified by HPLC with a C18 analytical column (Vydac 218TP 4.6 × 250 mm) using solvent A (water + 0.01% TFA) and solvent B (acetonitrile + 0.1% TFA); with a gradient from 0% to 60% of solvent B from minute 5 to minute 65 with a flow of 1 mL/min. The purity of the peptide was determined by mass spectrometry analysis. The peptide was vacuum dried and stored at −20 °C until use.

For the expression of MCP-1, the plasmid pQE30:MCP-1 was designed and constructed. The amino acid sequence is MRGSHHHHHHGSENLYFQGQPDAINAPVTCCYNFTNRKISVQRLASYRRITSSKCPKEAVIFKTIVAKEICADPKQKWVQDSMDHLDKQTQTPKT. The plasmid was used to transform *E. coli* Origami cells. One colony of the transformed culture was used to inoculate 50 mL of LB medium + ampicillin (100 μg/mL) and incubated at 37 °C for 16 h with shaking. 10 mL of the previous culture was taken and added to 1 L of LB medium + ampicillin (100 μg/mL); this was incubated at 37 °C and 200 rpm until an OD_600 nm_ of 0.8–0.9. 0.5 mM of IPTG was used as the inducer, and the culture was incubated at 16 °C for 18 h and 200 rpm.

After induction, cells were centrifuged at 5500× *g* for 15 min at 4 °C in a Beckman Coulter Avanti J-30I centrifuge. The resulting pellet was resuspended in 0.05 M Tris-HCl pH 8.0 buffer and passed through a French press at a pressure of 30 kpsi. The cell lysate was centrifuged at 10,000× *g* for 15 min (Beckman Coulter Avanti J-30I^®^, JA-14 rotor). The supernatant and the pellet were run on SDS-PAGE to detect the chemokine.

Protein recovered in the soluble fraction was purified by Ni-NTA resin affinity chromatography (QiaGen^®^). The protein bound to the solid phase was eluted with 50 mM Tris-HCl buffer + 400 mM imidazole, pH 8.0. Fractions of 1 mL were collected for analysis on 15% SDS-PAGE to verify purification. Finally, MCP-1 was purified through reversed-phase HPLC, with an analytical C4 column (Vydac 214TP 4.6 × 250 mm) using solvent A (water + 0.01% TFA) and solvent B (acetonitrile + 0.1% TFA); with a gradient from 0% to 60% solvent B from minute 5 to minute 65 and a flow rate of 1 mL/min. The MCP-1 was vacuum dried and stored at −20 °C until use.

Synthetic Css54 and the recombinant MCP-1 (2–3 µg) were reconstituted in 20 µL of 60% acetonitrile with 0.1% acetic acid and directly applied to a Thermo Scientific LCQ Fleet ion trap mass spectrometer (San José, CA, USA) using a Surveyor MS syringe pump delivery system. The eluate at 10 mL/min was split out to introduce only 5% of the sample into the nanospray source (0.5 mL/min). The spray voltage was set from 1.5 kV, and the capillary temperature was set to 150 °C. The fragmentation source was operated at 25–35 V of collision energy, 35–45% (arbitrary units) of normalized collision energy, and the scan with a wide band was activated. All spectra were obtained in the positive-ion mode. The data acquisition and deconvolution were performed on the Xcalibur Windows NT PC data system. The average molecular masses values vary about ±1 Da due to the limited resolution of this instrument. N-terminal Edman degradation was performed on a Shimadzu PPSQ-31A (Shimadzu, Kyoto, Japan) automated gas-phase sequencer. Sample (60 µg) was dissolved in 10 mL of 37% CH_3_CN (*v*/*v*) solution and applied to TFA-treated glass fiber membranes, pre-cycled with Polybrene (Sigma-Aldrich Co., St. Louis, MO, USA).

### 5.3. Determination of Minimum Inhibitory Concentration

The antimicrobial activity of the Css54 peptide and MCP-1 chemokine was determined by a serial microdilution test following the Clinical Laboratory Standard Institute criteria. For this, either Css54 or MCP-1 were diluted in Mueller–Hinton (MH) broth (50 µL) and incubated together with a bacterial suspension of *S.* Typhimurium (≈1 × 10^8^ bacteria/mL). The mixture was incubated at 37 °C for 18 h. Subsequently, the absorbance was measured at 595 nm. The lowest concentration at which the molecule inhibited the cell growth indicated the minimum inhibitory concentration (MIC). Ceftriaxone (100 µg/mL) was used as positive control and the bacterial culture without treatment was the negative control. MH broth and either Css54 or MCP-1 samples were also incubated to verify their sterility.

### 5.4. Minimum Bactericidal Concentration

A minimum bactericidal concentration (MBC) test was performed to determine whether Css54 or MCP-1 were bactericidal or bacteriostatic. The test provides the concentration at which the growth of 99.9% of the bacteria in the medium was inhibited. The starting point depended on the result of the MIC. For this, 10 μL was taken from the wells where growth was inhibited and seeded on MH agar. The plate was incubated at 37 °C for 24 h. The MBC was the minimum concentration that inhibited bacterial growth; if the result was less than four times the MIC value, the molecule was considered bactericidal [18,21].

### 5.5. Cytotoxicity

Using the CellTiter 96^®^ AQueous One Solution colorimetric reagent, the possible toxicity of the molecules was determined in RAW 264.7 cells. The solution contained a tetrazolium compound [3-(4,5-dimethylthiazol-2-yl)-5-(3-carboxymethoxyphenyl)-2-(4-sulfophenyl)-2H-tetrazolium, MTS] and an electron coupler (Etosultafo phenazine, PES). When MTS was reduced in the culture medium by living cells to the soluble product formazan, absorbance at 490 nm was considered proportional to the percentage of living cells. Thus, ≈1 × 10^5^ cells were incubated in 24-well plates in 600 μL of RPMI medium +10% SFBi for 24 h at 37 °C and 5% CO_2_. Subsequently, the medium was removed, and the new medium was supplemented with different concentrations of Css54 and MCP-1 (0.78–100 μg/mL) and incubated for 1 h. After this, the medium was removed, and the cells were washed with cold PBS. Again, 580 µL of fresh medium and 20 µL of CellTiter 96^®^ was added, and the plate was incubated for 1 h under the same conditions. The absorbance of the supernatant was measured at 490 nm.

### 5.6. Phagocytosis Stimulation

It was analyzed whether Css54 or MCP-1 increase the ability of macrophages to phagocytose bacteria. For this, ≈1 × 10^5^ RAW 264.7 cells were seeded in a 24-well plate with 600 μL of RPMI medium +10% SFBi and incubated for 24 h at 37 °C and 5% CO_2_. Afterward, different concentrations of Css54 and MCP-1 (from 0.78 to 12.5 μg/mL) were added, and cells were incubated for 18 h. Later, the supernatant was removed, and cells were washed with PBS; then, 1 × 10^7^ cells of *S.* Typhimurium were added to 600 µL of medium and incubated for 30 min. The supernatant was discarded, and the plate was washed with 300 μL of ceftriaxone (6 µg/mL) dissolved in RPMI + SFBi to eliminate the bacteria present outside the cells. Later, 400 μL of 0.5% Triton was added to lyse the cells, samples were centrifuged at 6000× *g* for 7 min, the supernatant was decanted, the pellet was resuspended in 1 mL of sterile PBS, and 20 μL was transferred to another well with 180 μL of PBS to make 20:200, 20:4000 dilutions. Measures of 10 μL of each dilution were seeded in triplicate on an XLD agar and incubated for 24 h at 37 °C. Subsequently, colony counting of non-ingested bacteria by macrophages was performed.

### 5.7. Neutralization of LPS of S. Typhimurium

The chromogenic endotoxin quantification assay (Thermo Fisher Scientific, Waltham, MA, USA) was performed to determine whether the molecules neutralized bacterial lipopolysaccharide (LPS). This test is based on the activation of the coagulation enzyme present in the limulus amebocyte lysate (LAL) when there is endotoxin, catalyzing the release of p-nitroaniline (pNA) present in the chromogenic substrate to produce a yellow color. Therefore, the color intensity is directly proportional to the amount of endotoxin present in the sample, the lower the absorbance, the greater the neutralization. From 0.18 to 6 μg/mL of either Css54 or MCP-1 was incubated with 0.5 EU/mL of *S.* Typhimurium LPS for 30 min at 37 °C. The control group only contained 0.5 EU/mL of LPS and water. The manufacturer’s instructions were followed, and absorbance at 405 nm was measured.

### 5.8. Sepsis Induction in Mice

The bioethics committee of the Biotechnology Institute of UNAM approved the protocol for the bacterial sepsis model with project number 371. It was performed under the requirements established by the Animal Housing Standards in our institution based on the Official Mexican Standard NOM 062-ZOO-1999. Male Balb/C mice of 20–23 g was used. They were separated into working groups (*n* = 3) under controlled conditions of light/dark 12:12 h, temperature, and humidity, with food and water ad libitum. Mice were injected with 4 × 10^6^ CFU of *S.* Typhimurium ATCC 14028 dissolved in 200 µL of PBS to induce infection. After 24 h, each group of mice received 15 µM intraperitoneally of Css54, MCP-1, or both, dissolved in PBS. Since both molecules differed in their molecular weight and length, we calculated the administration dose based on molarity.

### 5.9. Bacterial Load Count

After 72 h of infection, mice were sacrificed by cervical dislocation, and peritoneal lavage was performed. A 1 mL measure of sterile PBS was injected into the peritoneum and collected in a sterile tube. The wash was diluted at 1:10 and 1:100, of which 10 µL was seeded in triplicate on XLD agar. The plate was incubated for 24 h at 37 °C, and a bacterial count was performed. Similarly, the liver and spleen were extracted from each mouse. The organs were macerated in 1-mL PBS and then diluted at 1:10, and 10 µL was seeded in XLD agar. It was incubated for 24 h at 37 °C, and a bacterial count was performed.

### 5.10. Cytokine Analysis by Flow Cytometry

Blood was extracted after 72 h of infection in a tube with EDTA and centrifuged at 200× *g* for 2 min to separate the plasma, which was analyzed by a cytometric bead matrix to detect IL-12p70, TNF-α, IFN-γ, MCP-1, IL-10, and IL-6 (CBA, Mouse inflammation Kit- Becton Dickinson). For this, 50 μL of the sample was mixed with 50 μL of capture beads, and 50 μL of antibody against PE (Phycoerythrin) were added and incubated for 2 h at room temperature. After that, 1 mL of wash buffer was added, and the mix was centrifuged at 200× *g* for 5 min. The supernatant was discarded, and 300 µL of wash buffer was added to resuspend the pellet. Finally, the samples were examined on an Attune NxT acoustic cytometer (Thermo Fisher Scientific, Waltham, MA, USA) with a flow rate of 100 μL/min. Data were analyzed with FlowJo software version 10 (FlowJo, LLC, Ashland, OR, USA).

### 5.11. Statistical Analysis

The GraphPad Prism software (GraphPad, San Diego, CA, USA) was used for data analysis. The data in bars were the mean with its standard deviation (±SD). In the case of weight analysis, in vivo bacterial count, and cytokine expression profiles, an ANOVA analysis of the treatments was performed against the corresponding control.

## Figures and Tables

**Figure 1 antibiotics-11-00607-f001:**
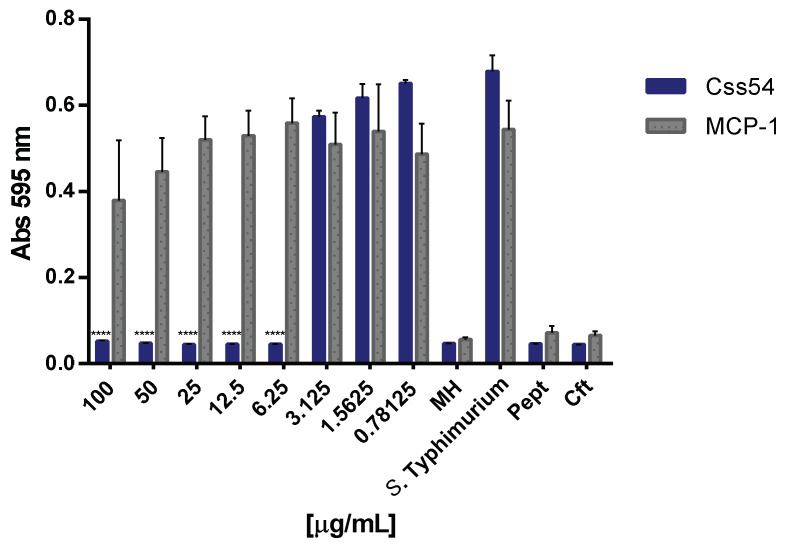
Css54 inhibits the growth of *Salmonella* Typhimurium. Different concentrations of Css54 and MCP-1 were incubated with 1 × 10^8^ cells of *S.* Typhimurium ATCC 14028 in Mueller–Hinton (MH) broth (*n* = 3). MH: broth without bacteria; *S*. Typhimurium: Bacteria without treatment; Pept: Css54 or MCP-1 [0.39 µg/mL each] + MH; Cft: bacteria + ceftriaxone [100 µg/mL]. The mean is the average of three independent experiments (**** *p* < 0.0001).

**Figure 2 antibiotics-11-00607-f002:**
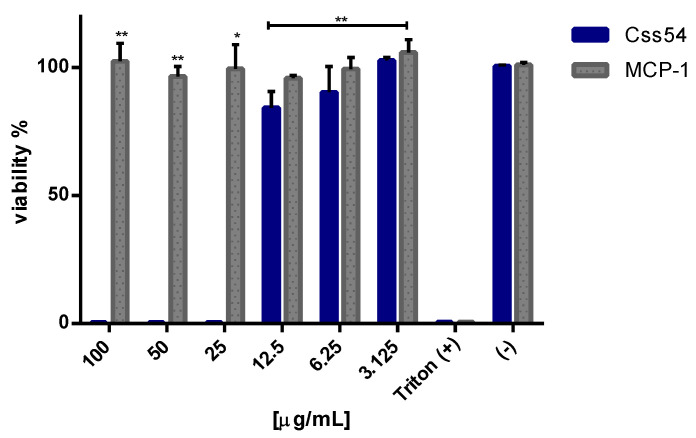
Css54 inhibits macrophage viability. Viability of RAW 264.7 (1 × 10^5^) cells challenged for one hour with different concentrations of Css54 or MCP-1 was determined. Triton (+): cells treated with 1% Triton and without Css54 or MCP-1; (−): cells without molecule. This figure shows the average of three independent experiments, (* *p* < 0.05; ** *p*< 0.01).

**Figure 3 antibiotics-11-00607-f003:**
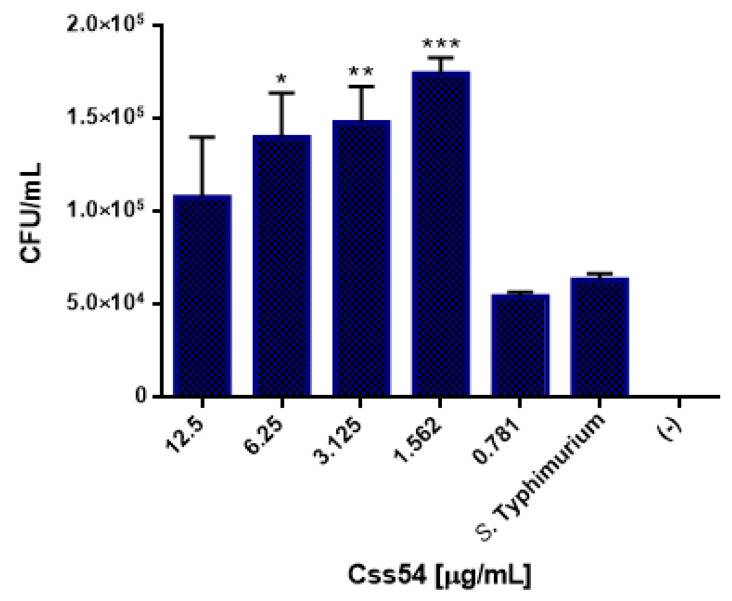
Css54 promotes phagocytic activity of RAW 264.7 cells. Macrophages were challenged with different concentrations of the peptide (0.78–12.5 μg/mL) for 18 h and infected with *S.* Typhimurium for 30 min to evaluate their phagocytic activity (*n* = 3). *S.* Typhimurium: infected cells without peptide; Control (−): cells without bacteria. Figure shows the average of three independent experiments (* *p* < 0.05; ** *p* < 0.01; *** *p* < 0.001).

**Figure 4 antibiotics-11-00607-f004:**
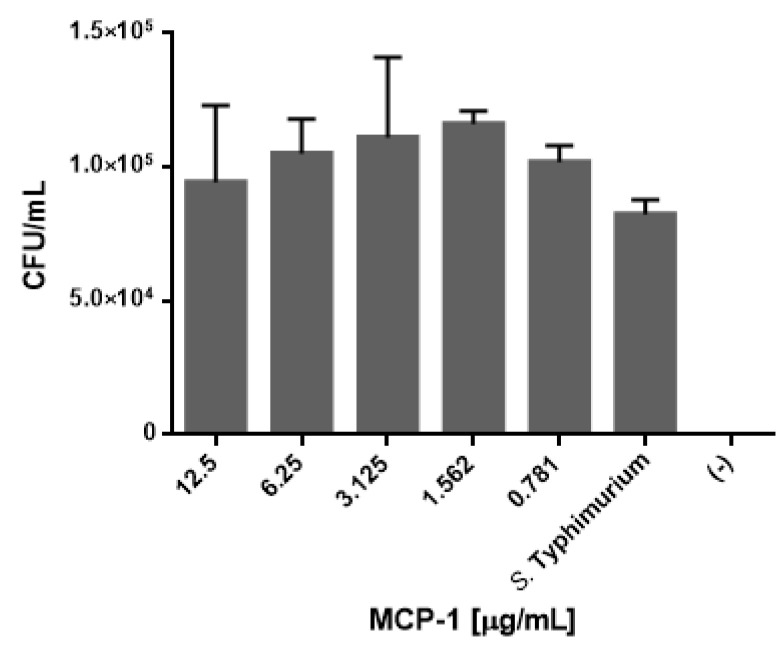
MCP-1 does not alter the phagocytic activity of RAW 264.7 cells. Macrophages were challenged with different concentrations of MCP-1 (0.78–12.5 μg/mL) for 18 h and infected with *S.* Typhimurium for 30 min to evaluate their phagocytic activity (*n* = 3). *S.* Typhimurium: infected cells without MCP-1; Control (−): cells without bacteria. This figure shows the average of three independent experiments (statistically non-significant).

**Figure 5 antibiotics-11-00607-f005:**
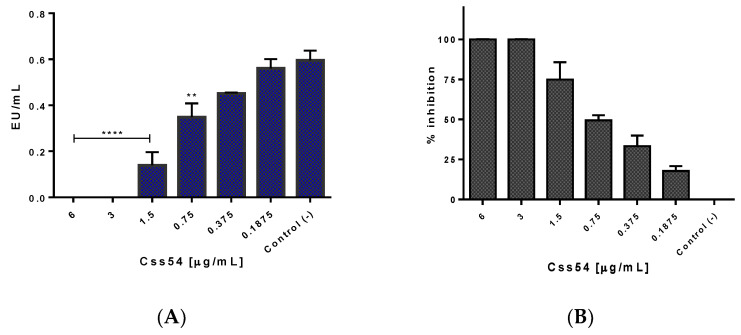
Neutralization of *S*. Typhimurium LPS by the Css54 peptide. Different concentrations (0.18–6 μg/mL) of Css54 were incubated with 0.5 EU/mL of *S.* Typhimurium LPS for 30 min. (**A**) Units of endotoxin (EU) per mL with respect to Css54 concentration; (**B**) Percentage of inhibition achieved by incubating the peptide with LPS. Control (−): LPS + water. This figure shows the average of three independent experiments, (** *p* < 0.005, **** *p* < 0.0001).

**Figure 6 antibiotics-11-00607-f006:**
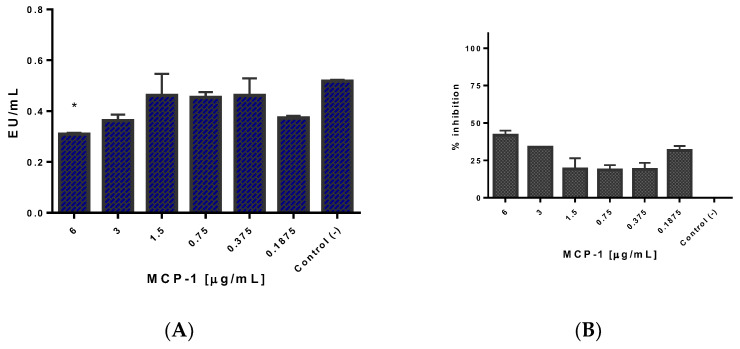
LPS from *S*. Typhimurium binds slightly to MCP-1. Concentrations of 0.18 to 6 µg/mL of MCP-1 were incubated with ≈0.5 EU/mL of *S.* Typhimurium LPS for 30 min (*n* = 3). (**A**) Endotoxin units (EU) per mL with respect to MCP-1 concentration; (**B**) Percentage of inhibition achieved by incubating the chemokine with LPS. Control (−): LPS + water. This figure shows the average of three independent experiments (* *p* < 0.05).

**Figure 7 antibiotics-11-00607-f007:**
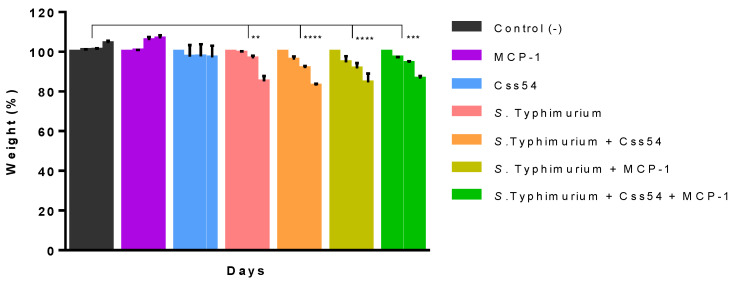
Weight control of mice with sepsis. Balb/C mice were infected with *S*. Typhimurium (≈4 × 10^6^ CFU) and treated with a single dose of Css54, MCP-1, and Css54 + MCP-1 (15 µM) at 24 h post-infection. Bars represent days 1 to 4 (72 h in total). Control (−): PBS (*n* = 6). This figure shows the average of six individuals from two independent experiments (** *p* < 0.01, *** *p* < 0.001, **** *p* < 0.0001).

**Figure 8 antibiotics-11-00607-f008:**
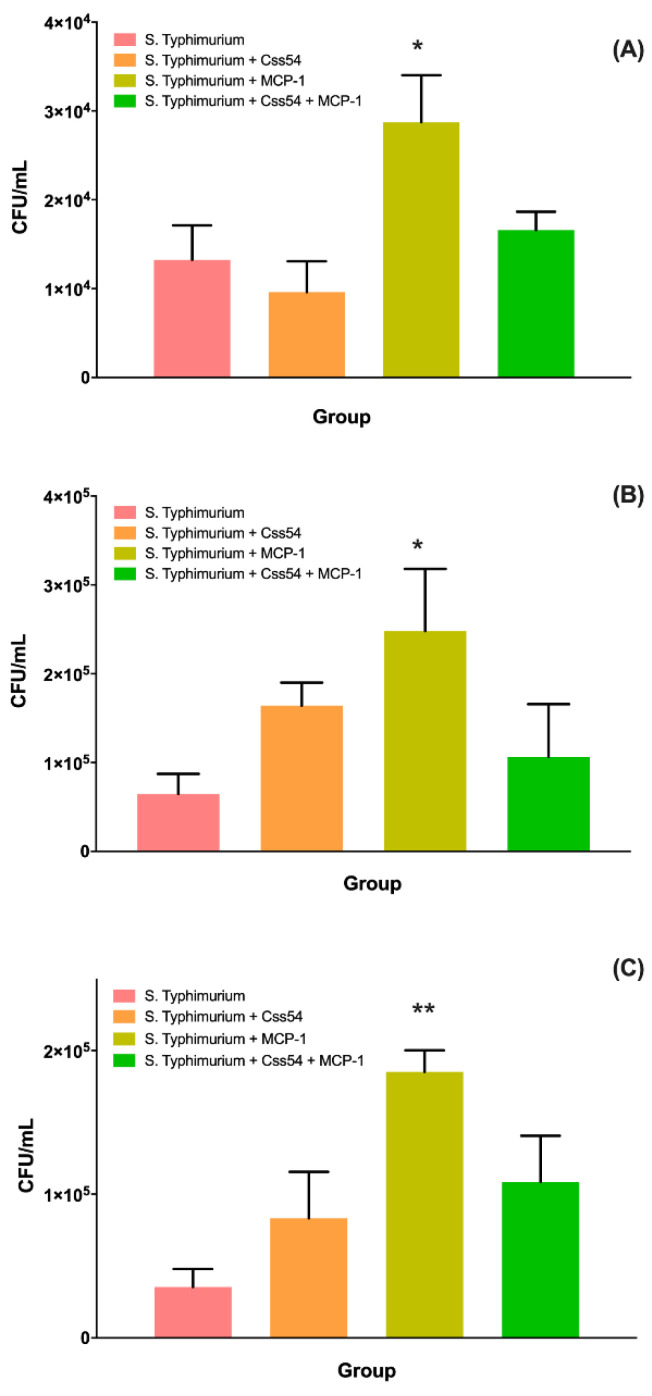
Css54 and MCP-1 did not decrease the bacterial load. Bacterial count in (**A**) peritoneum; (**B**) liver, and (**C**) spleen. Balb/C mice were infected with *S.* Typhimurium (≈4 × 10^6^ CFU) and treated with Css54, MCP-1, and Css54 + MCP-1 Css54 (15 μM) after 24 h of infection. After receiving the treatment, the status of the mice was monitored, and after 24 h, they were sacrificed to perform the bacterial count. The bacterial count was performed in the peritoneum at 72 h (*n* = 6). This figure shows the average and standard deviation of six individuals from two independent experiments (* *p* < 0.05; ** *p* < 0.01).

**Figure 9 antibiotics-11-00607-f009:**
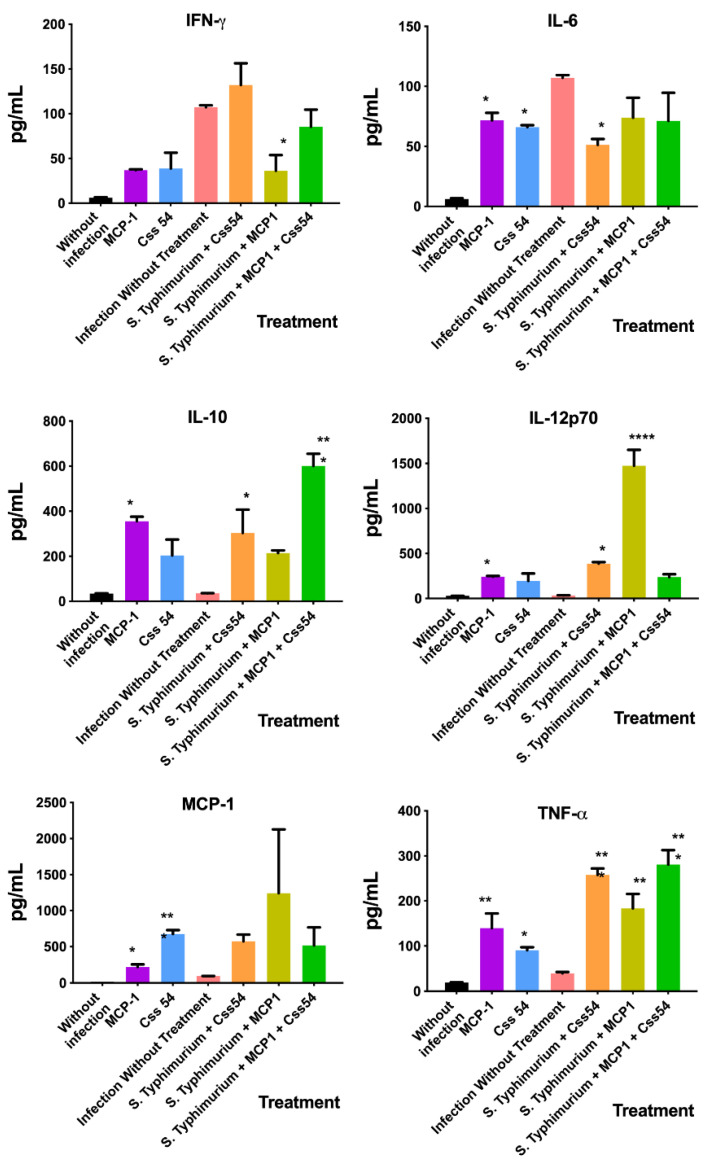
Cytokine expression was altered in mice treated with Css54 and MCP-1. Plasma from Balb/C mice after infection and treatment with Css54, MCP-1, and MCP-1 + Css54 was analyzed by flow cytometry at 100 μL/min. Css54, MCP-1, and MCP-1 + Css54: 15 µM; Infection without treatment: PBS. This figure shows the average and standard deviation of triplicates from two independent experiments (* *p* < 0.05; ** *p* < 0.01; **** *p* < 0.0001).

## Data Availability

Not applicable.

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
