# Peer review of "Antimicrobial and Immunomodulatory Effects of Selected Chemokine and Antimicrobial Peptide on Cytokine Profile during Salmonella Typhimurium Infection in Mouse"

_antibiotics, 2022, doi:10.3390/antibiotics11050607_

Round 1

Reviewer 1 Report

In this work authors studied antimicrobial peptide Css54 in combination with MCP-1 chemokine in the Salmonella Typhimurim sepsis model. Such works are of great interest in the search for new methods of treatment of sepsis.

In my opinion, the work can be accepted for publication after minor revisions. 

-- line 60: is this phrase necessary: “On the other hand”?

-- line 69: maybe it has to be like that: “…It has an antimicrobial activity against Gram-positive…”

-- line 177: since it is an animal experiment it would be better to show also exact concentrations of Css54 and MCP-1. For example, you can show mg per kg or mg per mice.

-- In section 2.6 I have not find the information on how often Css54 and MCP-1 were administered: every day or it was single administration after 24 h post-infection? I would recommend including this information in 2.6 section.

-- Figure 9 – the word “condition” – what does it mean?

Author Response

In this work authors studied antimicrobial peptide Css54 in combination with MCP-1 chemokine in the Salmonella Typhimurim sepsis model. Such works are of great interest in the search for new methods of treatment of sepsis.

In my opinion, the work can be accepted for publication after minor revisions. 
-- line 60: is this phrase necessary: “On the other hand”?

Answer: Thanks, now we have changed the sentence.

-- line 69: maybe it has to be like that: “…It has an antimicrobial activity against Gram-positive…”

            Answer: Thanks, we have made the change.

-- line 177: since it is an animal experiment it would be better to show also exact concentrations of Css54 and MCP-1. For example, you can show mg per kg or mg per mice.

Answer: Thanks for your comment, the dose of MCP-1 was 0.6mg/kg and the one of Css54 was 0.016mg/kg, but since both peptides are quite different in molecular mass and length, we have decided to calculate its administration dose based on their molarity. We have added this information in lines 467-469.

-- In section 2.6 I have not find the information on how often Css54 and MCP-1 were administered: every day or it was single administration after 24 h post-infection? I would recommend including this information in 2.6 section.

Answer: Thanks, we have added such information. It was “a single dose administration” for each treatment.

-- Figure 9 – the word “condition” – what does it mean?

Answer: Thanks, it refers to “Treatment” we have change to it.

Reviewer 2 Report

Dear authors, 

This is a quite comprehensive study. Congratulations. 

Sincerely

Author Response

This is a quite comprehensive study. Congratulations. 

Sincerely

Answer: Thank you very much.

Reviewer 3 Report

The study deals with the evaluation of an antimicrobial peptide Css54 and the chemokine MCP-1 as host defense peptides in infections with a Salmonella strain. While Css54 shows some in vitro effects on bactericidal activity but also cytotoxicity, MCP-1 does not, which is not surprising as it is not known for being antimicrobial. Also, in terms of destroying bacterial cell wall, only Css54 shows effects. However, when it comes to in vivo studies, both peptides fail to inhibit or decrease sepsis, either individually or in combination. However, they both show differing, but not very conclusive effects on different cytokines.

The study shows a decent amount of well done laboratory work. Experiments are well performed with sufficient data to make statistical analysis feasible and all controls are done. However, I feel that the manuscript needs some additional work in order to become publishable. First of all, it seems very random to me to choose Css54 and MCP-1. Why only MCP-1, why not going for other chemokines, too? What is the overall motivation for this study? Also, testing MCP-1 for bactericidal activity is not a nogo, but it does not come to my mind as the first thing to do, as it is no typical antimicrobial. And in fact, the studies showed exactly that. In this regard, the authors should also step back from any conclusions, that MCP-1 has a tendency to show typical antimicrobial activity like LPS-binding or alterations in phagocytic activity. Here, the authors also show differing opinions. While in the abstract and parts of the results clearly no antibacterial effects of MCP-1 are stated, in the discussion there is. Authors should not contradict themselves. What is really a pity is that using both peptides in vivo does not show any effects on sepsis, in fact, MCP-1 increased the concentration of bacteria in certain tissues. At least Css54 should show an effect, especially if injected intraperitoneally. With this regard, the title is a bit overenthusiastic, as there is no evidence that both peptides decrease inflammation in sepsis. And also, the data on effects on inflammatory proteins does not give conclusive results to explain the aforementioned results. This leaves me puzzled if the data are sufficient for publication. I would prefer to see some more data on this, and maybe the authors should concentrate only on Css54 for that, as MCP-1 did not show any beneficial effects, neither in vitro nor in vivo:

Check the effect of Css54 at varying concentrations injected into mice to see, if there is a dose-dependent effect on sepsis.

If you see effects in this study, also look into the excretion of chemokines and see, if this is dose-dependent. And then try to group the chemokines into pro- and anti-inflammatory to get a clear picture, what effects are regulated by Css54.

Make a stability investigation of Css54 in blood serum, intraperitoneal fluid or at least in cell supernatant to see how fast it is degraded.

Think about stabilizing Css54 and performing the studies again with the stabilized variant.

In any case, what would be a good control, is to use a scrambled Css54 sequence to make sure, the effects you see are really depending on the correct sequence.

Regarding analytics of the material the authors used, I have also some issues:

Please provide mass spec and HPLC chromatograms of the purified Css54. Check in your mass spec, if the monoisotopic peak has been labeled correctly. Otherwise you would be 1 Da off your calculated mass hinting at incorrect incorporation of one amino acid (e. g. glutamic acid instead of glutamine). Also, give some information on the mass spectrometer you used for analysis, especially in terms of the resolution it can provide. If it can resolve spectra down to isotope separation, you may want to look into the MCP-1 sequence by tryptic digestion and peptide-mass-fingerprinting to make sure it is correct.

When writing the manuscript, be as clear as possible regarding the motivation of your work. In the discussion, go into detail about the cytokines you looked into in your in vivo studies. Why did you choose these, what is known about them from other studies with antimicrobials and what does it tell you with respect to your study. Don’t be too short on this complicated section, because otherwise no clear picture will form. And also do not try to interpret things from your data that are simply not shown by them and don’t speculate.

I reject the paper but am willing to review a new submission of the manuscript with all the above mentioned issues taken into consideration.

Author Response

The study deals with the evaluation of an antimicrobial peptide Css54 and the chemokine MCP-1 as host defense peptides in infections with a Salmonella strain. While Css54 shows some in vitro effects on bactericidal activity but also cytotoxicity, MCP-1 does not, which is not surprising as it is not known for being antimicrobial. Also, in terms of destroying bacterial cell wall, only Css54 shows effects. However, when it comes to in vivo studies, both peptides fail to inhibit or decrease sepsis, either individually or in combination. However, they both show differing, but not very conclusive effects on different cytokines.

The study shows a decent amount of well done laboratory work. Experiments are well performed with sufficient data to make statistical analysis feasible and all controls are done.

However, I feel that the manuscript needs some additional work in order to become publishable.

First of all, it seems very random to me to choose Css54 and MCP-1. Why only MCP-1, why not going for other chemokines, too?

Answer: Thanks for your observations, we have added this information to clarify why we were interested whether MCP-1 had an antimicrobial effect. Lines 80-84.

Several reports have mentioned the importance of MCP-1 in the resolution of wound infections and sepsis despite of its lack of antimicrobial effect. This effect is often attributed to its role as host-defense peptide since it has a strong effect on the modulation of immune response and chemotaxis of immune response cells.

What is the overall motivation for this study?

Answer: Thanks, we have added this comment on the introduction section. Lines 85-91

Some studies have shown that the combination of antimicrobial peptides with host defense peptides or antibiotics are beneficial on the treatment for some infections. Here, we evaluated the use of the antimicrobial peptide Css54 together with the chemokine, MCP-1, as a treatment against sepsis caused by S. Typhimurium to evaluate if the combination of the antimicrobial effect of Css54 and the immune characteristics of MCP-1 are enough for resolve the infection in a murine model; and thus, propose a treatment for these infections.

Also, testing MCP-1 for bactericidal activity is not a nogo, but it does not come to my mind as the first thing to do, as it is no typical antimicrobial. And in fact, the studies showed exactly that. In this regard, the authors should also step back from any conclusions, that MCP-1 has a tendency to show typical antimicrobial activity like LPS-binding or alterations in phagocytic activity.

Answer: Thanks for your comment, we have added this information at the discussion section. Lines 284-290

Certainly, we mentioned LPS binding and some interference on bacterial growth as an apparent quality of MCP-1, even that this it is not its principal function. This is important because MCP-1 shows that can reduce the growth of S. Typhimurium albeit it does not inhibit it, the same thing goes to the LPS-binding effect. We might perform some for larger periods of time or increase the concentration of the chemokine and determine if this improves such effects.

Here, the authors also show differing opinions. While in the abstract and parts of the results clearly no antibacterial effects of MCP-1 are stated, in the discussion there is. Authors should not contradict themselves.

Answer: Thanks, you are correct. Now, we have changed the open sentence in the discussion section. 232-234

In this work, it was determined the antimicrobial and immunomodulatory activity of peptide Css54 and the effect on the immune response of MCP-1 in a model of infection bye Salmonella Typhimurium.

What is really a pity is that using both peptides in vivo does not show any effects on sepsis, in fact, MCP-1 increased the concentration of bacteria in certain tissues. At least Css54 should show an effect, especially if injected intraperitoneally.

With this regard, the title is a bit overenthusiastic, as there is no evidence that both peptides decrease inflammation in sepsis.

Answer: We consider this tittle due to the increase of anti-inflammatory cytokine IL-10. We have added a comment in lines 322-328.

It is worth to mention that when both peptides are applied in combination, there is a clearly increase in the expression of IL-10 and TNF-a, while others cytokines remains at the same level that those present in the infection group, which indirectly mean a tendency to keep controlled the inflammatory response but, at the same time, to create an anti-inflammatory environment; however, this aspect must be corroborated with the screening of more cytokines, inflammatory molecules and cell response differentiation.

And also, the data on effects on inflammatory proteins does not give conclusive results to explain the aforementioned results. This leaves me puzzled if the data are sufficient for publication. I would prefer to see some more data on this, and maybe the authors should concentrate only on Css54 for that, as MCP-1 did not show any beneficial effects, neither in vitro nor in vivo:

Answer: Certainly MCP-1 seems to have few relevance in the treatment of this infections, but it has effects on the expression of IFN-γ, IL-12p70 and TNF-α, which be according to its role as host defense peptide. We have added this comment on the discussion section. Lines 318-322

Curiously, even though MCP-1 does not act as an antimicrobial, it increases the expression of IL-12p70 and TNF-a, while decreases that of IFN-γ. There is also a non-significant increase in the expression of IL-10 by this chemokine, and a non-significant reduction of IL-6, which it may grant to think in further experiments by ELISA that could be helpful to clarify this modulation in the liberation of cytokines.

Check the effect of Css54 at varying concentrations injected into mice to see, if there is a dose-dependent effect on sepsis.

Answer: This concentration was stablished following the in vitro antimicrobial and cytotoxic effects. If were used higher concentrations it might be toxic to mice, and if we were used less amounts, we were expected nonantimicrobial effect.

If you see effects in this study, also look into the excretion of chemokines and see, if this is dose-dependent. And then try to group the chemokines into pro- and anti-inflammatory to get a clear picture, what effects are regulated by Css54.

Answer: Part of our interest is to analyze more cytokines and possibly signaling pathways, we just got a cytokines bead array kit that could analyze 17 cytokines, but we are currently within the standardization process. We hope that these results could shed light in another communication.

Make a stability investigation of Css54 in blood serum, intraperitoneal fluid or at least in cell supernatant to see how fast it is degraded.

Answer: Thanks, this is type of peptide blood serum kinetics are in our mid, for now we have introduced the following comments in the discussion section. Lines 299-302

A way to evaluate this, it could be implementing peptide blood serum kinetics or analyzing their biodistribution in a murine model. This data may be important for designing better treatments thru modifying the dose of peptides injected or using excipients during administration.

Think about stabilizing Css54 and performing the studies again with the stabilized variant.

Answer: Thanks for your comment, certainly, we have interest in developing antimicrobial structures and testing them in vivo following the spirit to find out if at least one of them could shed light by improving sepsis resolution. In this tone, we do not discard the chemical modification of peptides, or their way of administration, but for now this is the model that we are working with.

In any case, what would be a good control, is to use a scrambled Css54 sequence to make sure, the effects you see are really depending on the correct sequence.

Answer: This is an interesting idea, and we could have it as a control. Also, we have used other antimicrobial peptides, as experimental controls, for comparison to Css54.

Regarding analytics of the material the authors used, I have also some issues:

Please provide mass spec and HPLC chromatograms of the purified Css54

Answer:  The mass spec and HPLC chromatograms of the purified Css54 are now in the supplementary section.

Check in your mass spec, if the monoisotopic peak has been labeled correctly. Otherwise you would be 1 Da off your calculated mass hinting at incorrect incorporation of one amino acid (e. g. glutamic acid instead of glutamine).

Answer: We compared the experimental molecular mass of 2,870.4 da, and it agrees with that of the theoretical one (FFGSLLSLGSKLLPSVFKLFQRKKE). Also, N-terminal analysis (10-12 aa) was used to verify Css54 and MCP-1 amino acid sequences. Also, the nucleotides in the plasmid coding for MCP-1 were verify by DNA sequencing.

Also, give some information on the mass spectrometer you used for analysis, especially in terms of the resolution it can provide. If it can resolve spectra down to isotope separation, you may want to look into the MCP-1 sequence by tryptic digestion and peptide-mass-fingerprinting to make sure it is correct.

Answer: Information of the mass spectrometer is now included.

When writing the manuscript, be as clear as possible regarding the motivation of your work.

In the discussion, go into detail about the cytokines you looked into in your in vivo studies. Why did you choose these, what is known about them from other studies with antimicrobials and what does it tell you with respect to your study.

Answer:  Thank you for your interest in improving this communication. We have added the following information. Lines 307 to 313

Here, we analyzed the effect of peptides Css54 and MCP-1 in the profile expression of the inflammatory cytokines IFN-γ, IL-6, IL-12p70, TNF-α, the antinflammatory cytokine IL-10 and the chemokine MCP-1. All these molecules have their own physiology role during the infection by S. Typhimurium. It is worth to mention that this bacterium also takes advantage of the liberation of pro- and anti- inflammatory cytokines in mammals to cause damage in the intestine and, at the same time, to evade the immune response.

Don’t be too short on this complicated section, because otherwise no clear picture will form.

Answer:  Thanks, we have improved the redaction of this section from lines 313 to 328

Css54 demonstrated the ability to reduce IL-6 levels and increased those of the an-ti-inflammatory cytokine IL-10, and the pro-inflammatory cytokines IL-12p70 and TNF- that are involved in the inflammatory response and associated to the expression of IFN- α, which expression is lightly increased; however, IFN-γ increase is not enough for being seen as significative, but certainly it has merits for further studies. Curiously, even though MCP-1 does not act as an antimicrobial, it increases the expression of IL-12p70 and TNF- α while decreases that of IFN-γ. There is also a non-significant increase in the expression of IL-10 by this chemokine, and a non-significant reduction of IL-6, which it may grant to think in further experiments by ELISA that could be helpful to clarify this modulation in the liberation of cytokines. It is worth to mention that when both peptides are applied in combination, there is a clearly increase in the expression of IL-10 and TNF-α while others cytokines remains at the same level that those present in the infection group, which in-directly mean a tendency to keep controlled the inflammatory response but, at the same time, to create an anti-inflammatory environment; however, this aspect must be corroborated with the screening of more cytokines, inflammatory molecules and cell response differentiation.

And also do not try to interpret things from your data that are simply not shown by them and don’t speculate.

I reject the paper but am willing to review a new submission of the manuscript with all the above mentioned issues taken into consideration.

Answer:  Thank very much for your criticism, that certainly will improve this manuscript.

Round 2

Reviewer 3 Report

The article is still not ripe for publication. Title and abstract still contain conclusions that simply cannot be drawn from the data. Nowhere is shown that the combination of Css54 and MCP-1 reduces the infection in the sepsis model. In fact it is rather the other way round: Css54 increases bacterial amount in some organs of the infected organism. Additionally, the authors repeatedly speak of host-defense peptide when they refer to MCP-1. But MCP-1 is by no means a host-defense peptide, it isn't even a peptide. It is a protein, more specifically a chemokine. It regulates immune responses, that is true, and there is nothing to say against testing its effect in combination with a real host-defense peptide, but then experiments would have to be conducted differently. So, to me the authors did not convincingly show why to use MCP-1 at all. In addition, important articles on the subject were not cited. There has already been a study on Css54 in Salmonella Typhimurium, also showing hemolysis data. From there, one could already conclude, that it is not the best choice for further use in antimicrobial treatment. In the end, the English needs severe reviewing. Some sentences are 5 lines long and don't make sense, spelling and wording is often wrong. Taken together, in the present form, being the second submission, I reject the paper. The authors may want to consider concentrating on Css54 in a sepsis model and leave out the MCP-1 part for publishing these results somewhere else.

Author Response

The article is still not ripe for publication. Title and abstract still contain conclusions that simply cannot be drawn from the data.

Answer: we have now rewritten the title “Antimicrobial and immunomodulatory effects of selected chemokine and antimicrobial peptide on cytokine profile during Salmonella Typhimurium infection in mouse”; and some sentences in the abstract were deleted pej. “It is proposed that the combination of both peptides could be of interest for future treatments against inflammation caused by sepsis”.

Nowhere is shown that the combination of Css54 and MCP-1 reduces the infection in the sepsis model.

Answer: it is correct; there is no reduction in bacterial counts. If there were any related sentences, they have been modified/deleted.

In fact, it is rather the other way round: Css54 increases bacterial amount in some organs of the infected organism. Additionally, the authors repeatedly speak of host-defense peptide when they refer to MCP-1. But MCP-1 is by no means a host-defense peptide, it isn't even a peptide. It is a protein, more specifically a chemokine.

Answer: it is correct, MCP-1 is not an HDP. If there was any sentence related, it has been modified/deleted.

It regulates immune responses, that is true, and there is nothing to say against testing its effect in combination with a real host-defense peptide, but then experiments would have to be conducted differently. So, to me the authors did not convincingly show why to use MCP-1 at all.

Answers: No all experiments must have a happy ending. Here, we try MCP-1 because it is a chemokine that could have an impact on sepsis and other infections, as other researchers have reported.

In addition, important articles on the subject were not cited. There has already been a study on Css54 in Salmonella Typhimurium, also showing hemolysis data. From there, one could already conclude, that it is not the best choice for further use in antimicrobial treatment.

Answer: Yes, Css54 is hemolytic; that is why we used concentrations below the hemolytic IC50 to be administered to mice. No mice died during treatment. Some related articles are now cited.

In the end, the English needs severe reviewing. Some sentences are 5 lines long and don't make sense, spelling and wording is often wrong. Taken together, in the present form, being the second submission, I reject the paper.

Answer: English has been reviewed.

The authors may want to consider concentrating on Css54 in a sepsis model and leave out the MCP-1 part for publishing these results somewhere else.

Answer: We do not want to leave MCP-1 out of this communication because the information here could be interesting for other readers. This is the first article using recombinant MCP-1 in in vitro and in vivo assays, and its immunomodulatory effects. This data could be valuable for researchers interested in this interesting chemokine.